# A new method to assess mesoscale contributions to meridional heat transport in the North Atlantic Ocean

Andrew Delman[1] and Tong Lee[1]

[1]Jet Propulsion Laboratory, California Institute of Technology, Pasadena, CA, USA

**Correspondence:** Andrew Delman (adelman@jpl.caltech.edu)

**Abstract.** The meridional heat transport (MHT) in the North Atlantic is critically important to climate variability and the global overturning circulation. A wide range of ocean processes contribute to North Atlantic MHT, ranging from basin-scale overturning and gyre motions to mesoscale instabilities (such as eddies). However, previous analyses of "eddy" MHT in the region have mostly focused on the contributions of time-variable velocity and temperature, rather than considering the association of MHT with distinct spatial scales within the basin. In this study, a zonal spatial-scale decomposition separates large-scale from mesoscale velocity and temperature contributions to MHT, in order to characterize the physical processes driving MHT. Using this approach, we found that the mesoscale contributions to the time mean and interannual/decadal (ID) variability of MHT in the latitude range 39°–45°N are larger than large-scale horizontal contributions, though smaller than the overturning contributions. Considering the 40° N transect as a case study, large-scale ID variability is mostly generated close to the western boundary. In contrast, most ID MHT variability associated with mesoscales originates in two distinct regions: a western boundary region (70°–60° W) associated with 1–4 year interannual variations, and an interior region (50°–35° W) associated with decadal variations. Surface eddy kinetic energy is not a reliable indicator of high MHT episodes, but the large-scale meridional temperature gradient is an important factor, by influencing the local temperature variance as well as the local correlation of velocity and temperature. Most of the mesoscale contribution to MHT at 40° N is associated with transient and propagating processes, but stationary mesoscale structures explain most of the mesoscale MHT south of the Gulf Stream separation, highlighting the differences between the temporal and spatial decomposition of meridional temperature fluxes.

## 1   Introduction

Meridional heat transport (MHT) is essential to both the mean and variability of global climate. The time-mean MHT in the ocean is substantially lower than that in the atmosphere in mid-latitudes (Trenberth and Caron, 2001). However, oceanic MHT variability is important to the time variability of heat transport in the Earth system, particularly on interannual and decadal timescales (e.g., Häkkinen, 1999). Oceanic MHT is primarily associated with several physical processes: (1) overturning circulations with zonal-mean flows in distinct depth ranges carrying waters of different temperatures, (2) gyre circulations

advecting waters of different temperatures at distinct longitudes, and (3) mesoscale dynamics including coherent vortices ("eddies") developing from flow instabilities, as well as mesoscale-intensified jets and recirculations that are sustained by nonlinear momentum advection and rectified eddy fluxes (e.g., Hoskins et al., 1983; Waterman and Hoskins, 2013; Delman et al., 2015). Due to the steep vertical temperature gradients in much of the ocean, the overturning contribution to MHT has received the most attention in observational analyses of MHT (e.g., Talley, 2003, for a discussion of overturning MHT contributions)

The Atlantic basin has attracted particular interest in studies of the oceanic MHT, because of the role of the Atlantic Meridional Overturning Circulation and its implications for regional and global climate. The MHT in the north Atlantic ocean has been estimated as the transport required to balance air-sea heat flux observations (Hsiung, 1985) and as the residual from the top-of-atmosphere radiation balance that is unexplained by atmospheric transport (Trenberth and Caron, 2001); however, these methods of quantifying MHT are only useful for time-mean MHT since the time-mean change in heat storage is small compared to radiative fluxes and heat transports. Estimates of MHT have also been derived from in-situ measurements by ships (e.g., Hall and Bryden, 1982, and references therein; Koltermann et al., 1999; Talley, 2003) and autonomous Argo floats (Hobbs and Willis, 2012). Moreover, oceanic MHT at 24–26.5°N has been estimated using observations from the Rapid Climate Change-Meridional Overturning Circulation and Heatflux Array (RAPID-MOCHA) of moorings (Johns et al., 2011). Yet explicit estimates of MHT from ocean observations are often based on sparsely distributed measurements, and are likely to underestimate the contribution of mesoscale dynamics. Johns et al. (2011) estimated an Atlantic "eddy" MHT contribution of 0.10±0.03 PW at 24°-26°N based on spatially covarying velocity and temperature in five hydrographic sections, which is small compared to the estimated ∼1.3 PW total MHT at these latitudes. However, ship-based measurements do not generally show the full scope of MHT variability at a given latitude, and in the Atlantic there is far more mesoscale activity where the Gulf Stream separates from the western boundary.

Ocean models, particularly high-resolution eddy-permitting general circulation models (GCM), have been valuable tools to estimate the time mean MHT as well as its variability. An important caveat is that most of these studies consider any deviation of velocity or temperature from time-mean values to be an "eddy" contribution to MHT. Using a 1/4° resolution ocean GCM, Jayne and Marotzke (2002) estimated the time mean northward eddy MHT in the North Atlantic to peak at approximately 0.1 PW near 40°N, a smaller contribution than they found in tropical basins. Despite this relatively small value, and the fact that the grid resolution 1/4° is insufficient to resolve the baroclinic deformation radius at 40°N (Hallberg, 2013), time-mean maps of temperature fluxes in the Jayne and Marotzke study display many mesoscale features. Using a higher-resolution 1/12° model, Tréguier et al. (2017) found somewhat higher time-mean values of the eddy heat flux near 40°N, with a sharper peak of approximately 0.3 PW at 36.6°N near the Gulf Stream separation. Volkov et al. (2008) used a state estimate with mean horizontal grid spacing of 18 km, and limited the definition of eddy heat flux to comprising velocity and temperature anomalies at timescales shorter than 3 months; they found much smaller time-mean eddy heat fluxes (near zero at 40°N), though with a temporal standard deviation slightly over 0.1 PW. The results of these studies imply that eddy heat fluxes are dependent not just on model resolution, but on how "eddies" are defined. A canonical eddy is typically a vortex that is no more than a few hundred kilometers in diameter (i.e., mesoscale), but model analyses have not yet to our knowledge quantified basin-integrated temperature fluxes according to the spatial scale of the processes driving the fluxes. Moreover, observational estimates that

use (remote sensing-based) eddy tracking methods to quantify eddy contributions to heat transport (e.g., Hausmann and Czaja, 2012; Dong et al., 2014; Sun et al., 2019) have reached divergent conclusions depending on the methodology and scope of what is considered part of the eddy transport. One way to address this ambiguity is to consider the impact of all mesoscale dynamics on meridional heat transport, whether or not these mesoscale fluxes are due to coherent vortices.

The focus of this study is to quantify the time-mean and time-variable contribution of mesoscale dynamics to meridional heat transport in the North Atlantic, with a particular focus at 40°N where the Gulf Stream extension is nearly zonal and eddy kinetic energy (EKE) is near its maximum. Section 2 will discuss the eddy-permitting model simulation and the framework used to decompose the temperature fluxes into overturning, large-scale, and mesoscale components. Section 3 summarizes the results and the cross-basin structure of mesoscale temperature fluxes, while Section 4 relates the variability of mesoscale temperature fluxes to other indicators of the ocean state such as EKE and meridional temperature gradient. Section 5 discusses novel aspects of the approach presented here compared to earlier formulations of eddy fluxes, while Section 6 summarizes the conclusions of this study.

## 2    Methods

### 2.1    Model simulation and data

Most of our analysis uses output from a numerical simulation of the Parallel Ocean Program (POP) 2, a primitive equation ocean model (Smith et al., 2010). POP is the ocean component of the Community Earth System Model framework, and in this simulation it was configured on a tripole grid with two north poles over Canada and Siberia—more details of the simulation are found in Johnson et al. (2016) and Delman et al. (2018). The simulation was run on the Yellowstone computing cluster (Computational and Information Systems Laboratory, 2016) with a resolution of 0.1° at the equator, and approximately 8 km in the mid-Atlantic; it was forced with Coordinated Ocean-Ice Reference Experiments version 2 (CORE.v2; Large and Yeager, 2009), an interannually-varying flux dataset based on the National Centers for Environmental Prediction reanalysis with corrections from various satellite data. The model integration was started from 15 years of spin-up using CORE normal-year forcing (Large and Yeager, 2004), and run over 33 years (corresponding to forcing for the years 1977–2009); our analysis encompasses 32 years and begins in 1978 to focus on the period after the post-spin up transition. State variables (including velocity and temperature) and temperature fluxes have been archived from this simulation in 5-day averages, facilitating studies of mesoscale processes which frequently vary on intraseasonal timescales.

In addition to the model simulation output, the ocean surface dynamic topography dataset merged from various altimeters is used to validate the model's mean state and its representation of ocean surface variability. This dataset is produced by Collecte Localisation Satellites (Ducet et al., 2000) and available through the Copernicus Marine Environment Monitoring Service (CMEMS) at 1/4° spatial and daily temporal resolution. Compared to the altimetry-based dynamic topography, the model reproduces most essential features of the circulation in the North Atlantic (Fig. 1), though there are several important issues that have been previously noted in other model simulations (for a review of these issues see Chassignet and Marshall, 2008). These include the Gulf Stream separating from the continental slope at 38°-39°N in the model (Fig. 1b) vs. ~36°N in

observations (Fig. 1a), and too low eddy kinetic energy in the Northwest Corner region near 50°N, 40°W and in the Azores Current region near 34°N (Fig. 1c,d). These inaccuracies have been resolved in regional simulations of the North Atlantic at 1/10° resolution (e.g., Bryan and Smith, 1998; Smith et al., 2000; Bryan et al., 2007), but persist in global simulations even at the same resolution (e.g., Maltrud and McClean, 2005; Kirtman et al., 2012; Griffies et al., 2015). Therefore the focus of this study is on 40°N, in between the Gulf Stream Separation and the Northwest Corner where the distribution of EKE in the model is qualitatively similar to observations (Fig. 1c,d). The zonally-averaged values of EKE at 40°N are lower in POP than in the altimetry data, but this is true of much of the North Atlantic, and the zonally-averaged EKE peaks in altimetry and in the model are both at latitudes near 40°N (Fig. 1e).

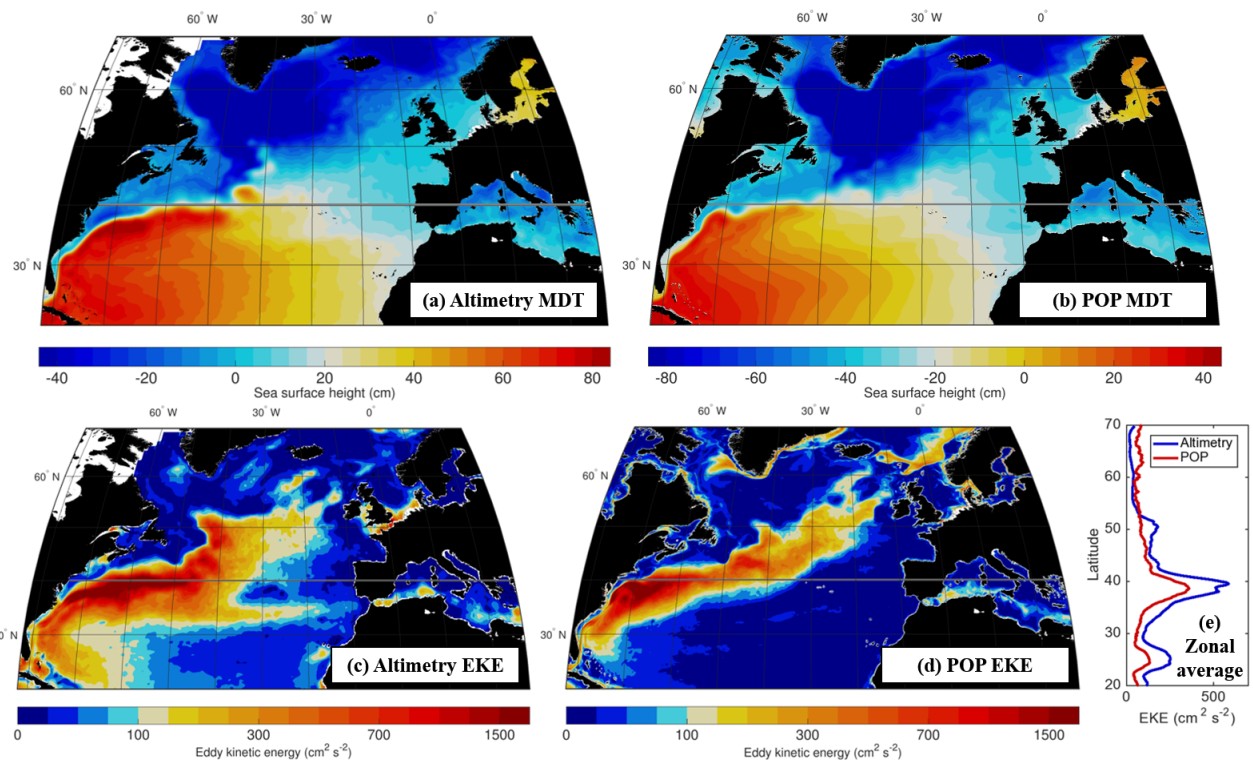

**Figure 1.** Comparison of mean dynamic topography from (a) CMEMS altimetry (1993-2016) and (b) POP simulation (1978-2009). The gray line indicates the 40°N latitude transect, the focus of this study. (c, d) Same as (c, d) but for surface eddy kinetic energy (EKE); POP sea surface height is low-pass filtered to remove variability at scales smaller than 0.5°, the Nyquist wavelength of the altimetry product. (e) Comparison of Atlantic zonally-averaged surface EKE from altimetry and POP.

## 2.2 Temperature flux decomposition

### 2.2.1 Previous decompositions

The meridional heat transport in the ocean is commonly regarded to consist of a mean and "eddy" component, where the "eddy" component is associated with velocity and temperature deviations from a temporal mean. Namely

$$\rho c_\rho \overline{vT} = \rho c_\rho \left( \overline{v}\,\overline{T} + \overline{v'T'} \right) \tag{1}$$

with $\rho$ the reference density and $c_\rho$ the specific heat capacity of seawater, and the meridional velocity and temperature are decomposed into time mean and deviation components $v = \overline{v} + v'$ and $T = \overline{T} + T'$ respectively. There are also cross-terms between the mean and deviation components in eq. (1), but these are zero by definition in time means. When the left-hand side is integrated zonally and vertically across a basin with zero net meridional flow ($\iint v\,dx\,dz = 0$), these heat fluxes are considered heat transports. The decomposition in eq. (1) is convenient for describing how much heat transport can be explained (and not explained) by the ocean's mean state, but it provides little information about the processes actually responsible for the heat transport. An alternative approach to this temporal decomposition is to use a depth and/or zonal average. Hall and Bryden (1982) first separated the flux $vT$ into depth-averaged $\overline{v}\,\overline{T}$ (barotropic) and deviations from this average $v'T'$ (baroclinic) contributions. Then the baroclinic component was further separated into zonal mean and zonal deviation terms.

### 2.2.2 Large-scale/mesoscale spatial decomposition

In this study we highlight the contributions of mesoscale dynamics, which may contribute to barotropic and baroclinic fluxes, but in a large ocean basin are always associated with deviations from the zonal mean. Hence we separate $v$ and $T$ into zonal mean and deviation components first; neglecting $\rho c_\rho$ to focus on the temperature flux, this decomposition is

$$\int vT\,dx = \int \left( \langle v \rangle \langle T \rangle + v''T'' \right) dx \tag{2}$$

where $\langle \, \rangle$ and $''$ indicate zonal mean and deviation respectively. While eq. (1) holds true only when the terms are time averaged, eq. (2) involves zonal integration or averaging, and therefore holds true without any time averaging. Since the term $\langle v \rangle \langle T \rangle$ varies in depth and time but not zonally, it quantifies the contribution to temperature transport from the overturning circulation, i.e., the meridional movement of warmer shallow waters over cooler deeper waters. The contributions of lateral variations in the ocean are therefore contained in the $v''T''$ term. However, these lateral variations include a wide range of processes that range in scale from subtropical and subpolar basin gyres, to instabilities at the smallest scales resolved by the model or observing system.

In order to separate the contribution of large-scale processes (e.g., gyres, long planetary waves) from mesoscale processes (e.g. transient and standing eddies), we introduce a further decomposition of the $v''T''$ term:

$$v''T'' = v_L T_L + [(v_L T_M + v_M T_L) + v_M T_M]$$ (3)

in which the subscripts $_L$ and $_M$ denote large-scale and mesoscale components of meridional velocity and temperature. The middle cross-terms on the right-hand side of eq. (3) are considered part of the mesoscale contribution since they would not exist without mesoscale processes. The separation of large-scale and mesoscale components is carried out in the spectral (wavenumber) domain as indicated in eq. (4)-(5). In order to preserve the large-scale volume transports in the structure of $v_L$, the meridional velocity $v$ is weighted (multiplied) by the lateral distance $\Delta x$ of each grid cell of the transect and then divided by the low-pass filtered $\Delta x$ after filtering. Given the zonally-detrended $v\Delta x$ and its associated Fourier wavenumber coefficients $V(k)$, the low-pass and high-pass transfer functions are

$$V_L(k) = \left[ 0.5 + 0.5\mathrm{erf}\left( -s\, ln\frac{|k|}{k_0} \right) \right] V(k)$$ (4)

$$V_M(k) = \left[ 0.5 + 0.5\mathrm{erf}\left( s\, ln\frac{|k|}{k_0} \right) \right] V(k)$$ (5)

and analogously for the decomposition of temperature (without the $\Delta x$ weighting). It can be readily seen that the sum of eq. (4) and eq. (5) is the original $V(k)$. After filtering, the zonal trends are added back to the large-scale velocity and temperature, but not the mesoscale velocity and temperature. Meridional velocity outside the basin boundaries (to a distance of $1/k_0$ from the outermost boundaries) and within interior land areas is set to zero; however, to minimize abrupt jumps in temperature and its zonal derivative at the boundaries, a buffer is also included at the western and eastern boundaries after detrending but prior to the application of the filters in eq. (4)–(5). Given $x_b$ the western boundary position and $x_c$ the zonal decorrelation scale of temperature at a given latitude, depth, and time, the average values of temperature in the ranges $x_b < x < x_b + x_c$ and $x_b + x_c < x < x_b + 2x_c$ ($T_1$ and $T_2$ respectively) are computed, and from these a boundary value $T_b = 1.5T_1 - 0.5T_2$ and slope $T_s = (T_2 - T_1)/x_c$. Then an error function is fitted outside the boundary that approximates the slope of the temperature profile approaching the western boundary

$$T''|_{x<x_b} = T_b \left\{ 1 + \mathrm{erf}\left[ \frac{\sqrt{\pi}}{2} \left| \frac{T_s}{T_b} \right| (x - x_b) \right] \right\}$$ (6)

and the mirror opposite formulation is applied to the eastern boundary. Where interior land areas wider than 1° longitude are present (e.g., between the main Atlantic basin and marginal seas), the zonal temperature profiles are separated into segments, and the low-pass filter in eq. (4) is applied to each segment. This low-passed temperature is used as a basis for interpolation across the land gap(s), and then the filters (4)–(5) are applied to the full temperature transect including the land gaps. This

procedure yielded the most credible results in regions of complex topography and sharp gradients, such as the Florida coast and in the Mediterranean basin.

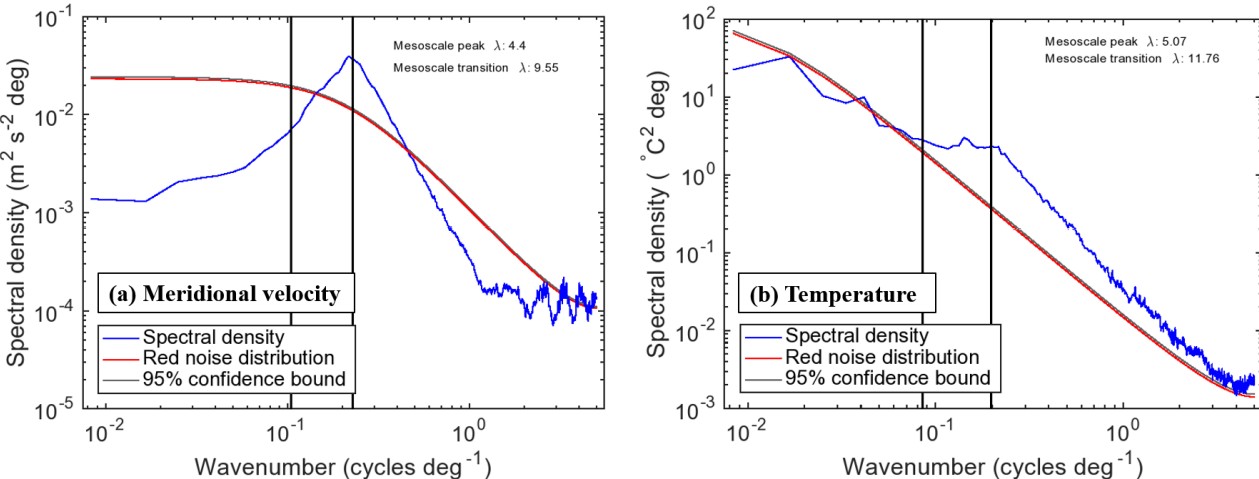

**Figure 2.** Zonal wavenumber spectral density estimate of (a) meridional velocity and (b) temperature in the Atlantic at $40°$N from POP. The vertical lines and text indicate the wavelength (in degrees) of the mesoscale transition and peak as calculated from each spectral estimate, based on the logarithmically-smoothed spectral profile. The red and gray curves indicate the red noise distribution expected from lag-1 (model grid-scale) autocorrelation and the corresponding 95% upper confidence bound.

There are two parameters in the transfer functions in eq. (4)–(5) that need to be chosen: $s$ the steepness factor at the wavenumber cutoff, which is set to 5, and $k_0$ the cutoff wavenumber. To determine a sensible value for $k_0$ and the related cutoff wavelength $\lambda_0 = 1/k_0$, spectral density estimates are computed from the POP meridional velocity $v$ and temperature $T$ fields at various latitudes in the Atlantic basin, with the results for $40°$N shown in Figure 2. With high levels of mesoscale activity at $40°$N, a clear mesoscale peak can be seen at wavenumbers corresponding to wavelengths between $4°$ and $5°$ longitude (Fig. 2a); there is a similar peak in wavenumber spectra of the Gulf Stream path through this region (e.g., Lee and Cornillon, 1996). The temperature spectral peak is less obvious, but is still visible relative to the general downward slope with increasing wavenumber (Fig. 2b). Wavelengths corresponding to the mesoscale peak and large-scale/mesoscale transition were identified by smoothing the spectral density curves for $v$ and $T$ in logarithmic space and identifying the minimum and maximum of $\partial^2 \left(\ln V\right)/\partial \left(\ln k\right)^2$ within broad expected ranges ($2°$-$20°$ wavelengths for the mesoscale peak, $3°$-$30°$ wavelengths for the large-scale/mesoscale transition). Though computed independently, wavelengths for the mesoscale peak and transition identified from the spectral curves of $v$ and $T$ are similar, providing confidence that the mesoscale signal starts to emerge at wavelengths shorter than $10°$ and peaks at approximately $5°$. Though a $5°$ wavelength is much longer than the first baroclinic Rossby radius ($\sim$20 km at this latitude), it does compare favorably with eddy radii of approximately 100 km observed at $40°$N in altimetry data (Chelton et al., 2011); the peak wavelength is expected to be approximately 4 times the typical eddy radius. Hence to obtain an optimal

separation between mesoscale and large-scale processes, the cutoff wavenumber in eq. (4) and (5) is set to $k_0 = 1/10$ cycles/°, corresponding to a cutoff wavelength of $\lambda_0 = 10°$.

### 2.2.3   Boundary and channel corrections to the meridional velocity

In order for the large-scale and mesoscale flux components to represent meaningful transports, a desirable attribute is that the

zonal integrals of $v_L$ and $v_M$ each sum to approximately zero across the transect. Moreover, the mesoscale velocity $v_M$ should generally have a zonal integral close to zero within the transect, across length scales greater than the cutoff wavelength $\lambda_0$. The filters in (4)–(5) satisfy these criteria for interior regions of the oceans, but where there are sharp spikes in $v$ near boundaries (i.e., wherever there is a strong boundary current) the $v_L$ and $v_M$ components may have large compensating zonal integrals unless corrections are applied.

The correction procedure is as follows: when the low-pass spatial filter (4) is applied, nonzero $v_L$ will bleed into land areas that are adjacent to a boundary; in order to conserve the large-scale structure of zonally-integrated $v$, this non-zero $v_L$ needs to be re-distributed over nearby water areas in the transect. For each point $x_{\text{ref}}$ over land, $v_L$ at $x_{\text{ref}}$ is re-distributed according to the triangular window function

$$w(x) = (1/\lambda_0)\,(\lambda_0 - |x - x_{\text{ref}}|)\,, \ \ \text{if } x \text{ is over water and } |x - x_{\text{ref}}| < \lambda_0$$

$$w(x) = 0, \qquad\qquad\qquad\quad \text{otherwise} \qquad\qquad\qquad\qquad\qquad\qquad\qquad\qquad\qquad\qquad (7)$$

and $w(x)$ is normalized so that its zonal integral from $x_{\text{ref}} - \lambda_0$ to $x_{\text{ref}} + \lambda_0$ is 1. If the land point is outside the westernmost or easternmost points of the entire basin, or within $\lambda_0/4$ of these points, then $w(x)$ is instead normalized so that it integrates to 1.1, to account for additional loss of meridional velocity from filtering at the edges. For an interior land region, the left and right sides of the triangular window are weighted according to the proximity of each boundary and the magnitude of $v_L$

at the boundary, with weighting factors $\sqrt{|x_{\text{bound}} - x_{\text{ref}}||v_L|}$ evaluated at the closest water point $x_{\text{bound}}$ on each side, and then normalized so that the zonal integral of $w(x)$ is 1 or 1.1. This weighting performs well at conserving the volume transport in segments on each side of narrow land areas. The correction applied to $v_L$ is then subtracted from $v_M$ so that the sum of the two components remains the same.

Another issue arises in narrow channels that contain strong currents (e.g., the Gulf Stream in the Florida Strait); in such cases

the channel is not wide enough to resolve a separation between the large-scale and mesoscale. In channels narrower than 2.5° longitude ($\lambda_0/4$), the $v_L$ is set equal to the total $v$ (minus the zonal mean), and $v_M$ is set to zero. Then any residual associated with the difference between the original $v_L$ and $v$ is redistributed using the triangular window in eq. (7), with channel areas as well as land areas in the window set to zero.

# 3 Variability and cross-basin structure of temperature fluxes

## 3.1 Flux decomposition variation with latitude

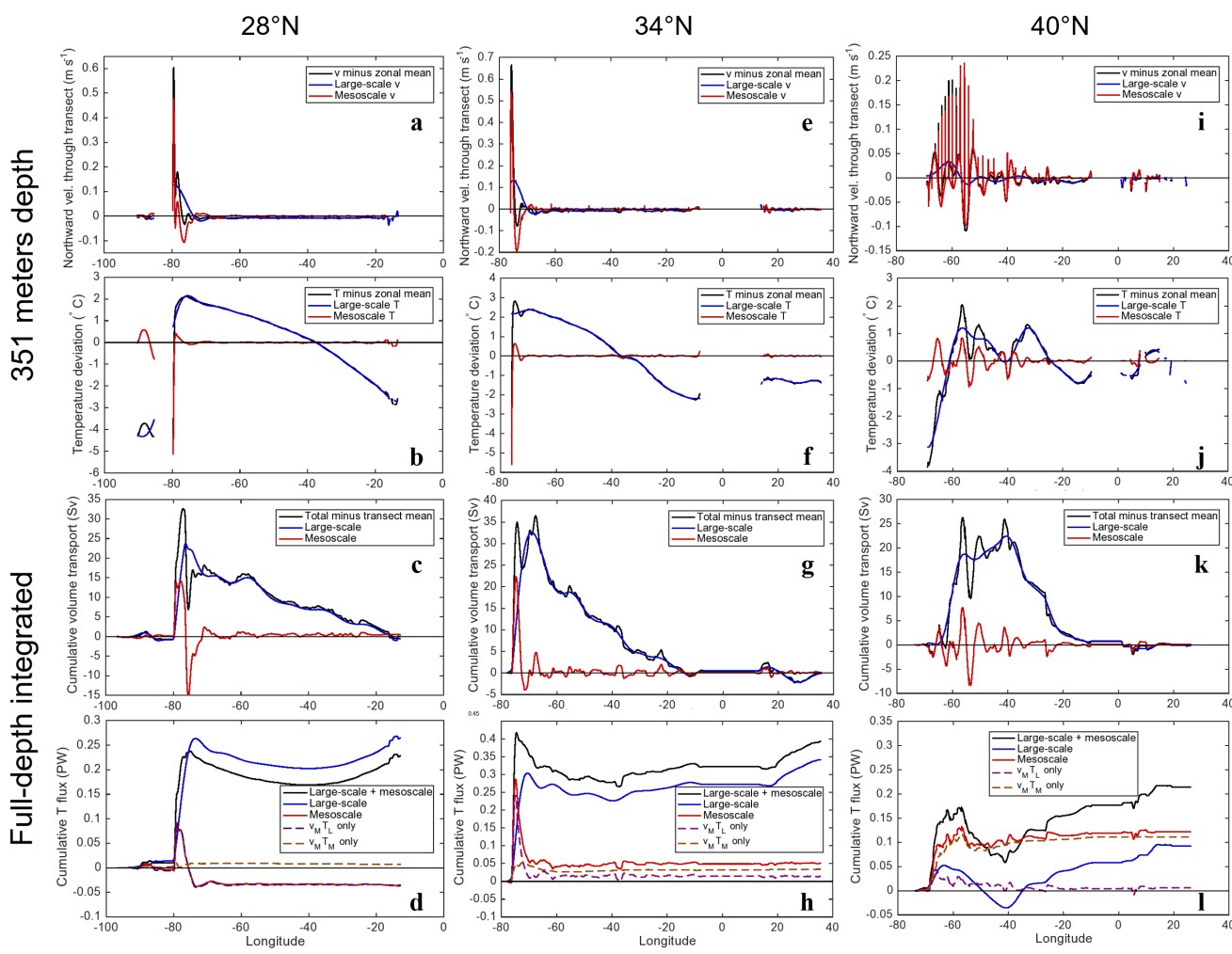

**Figure 3.** Structure of the time mean velocity and temperature spatial decomposition at 351 m depth for various latitudes, along with the full-depth cumulative zonally-integrated volume transport and temperature flux. The left column shows the (a) northward velocity through the transect line and (b) temperature spatial decomposition at 351 m depth for 28°N, along with the time-mean full-depth cumulative (c) volume transport and (d) non-overturning temperature flux. The center and right columns are the same but for latitudes (e)-(h) 34°N and (i)-(l) 40°N respectively.

Applying the procedure described in Section 2.2 at several different latitudes, it is possible to obtain an understanding of the spatial scales associated with temperature flux contributions (Figure 3). For example, contrasts can be observed in the non-

overturning MHT in the Atlantic at 28°N and 34°N, both latitudes at which a strong northward western boundary current is compensating a broad interior southward Sverdrup flow (Fig. 3a–h). Most of the non-overturning temperature flux is associated with the large-scale component at both latitudes; however, the mesoscale temperature flux (MTF) switches sign from negative at 28°N to positive at 34°N (Fig. 3d,h). The reason for this is the temperature difference between the core of the northward boundary current and the southward recirculation ∼2° to the east. At 351 m depth (a representative depth for lateral temperature gradients in the thermocline), the temperature at 28°N is lower in the boundary current than it is in the interior recirculation (Fig. 3b), as isopycnals tilt upward sharply approaching the Florida coast; this large-scale temperature gradient interacts with the mesoscale velocity structure and therefore the $v_M T_L$ term explains the negative MTF (Fig. 3d). However, at 34°N the temperature peak along the zonal profile is coincident with the boundary current (Fig. 3f), and the temperature peak also has more of a mesoscale signature that explains why the $v_M T_M$ contributes the most to the MTF (Fig. 3h). (The $v_L T_M$ term is typically negligible, owing to the red-shifted spectra of temperature relative to velocity as shown in Figure 2.) At 40°N, the angle of the grid combined with a large zonal current in the western part of the basin results in large spikes in the velocity field (Fig. 3i); fortunately the decomposition method still retains the large-scale structure of the volume transport (barotropic streamfunction) in the large-scale velocity component (Fig. 3k). Notably, the time-mean contribution of the MTF is much larger at 40°N, though most of this contribution also occurs near the western side of the basin (Fig. 3l).

The contributions of the overturning, large-scale, and mesoscale components to basin-integrated heat transport can be computed for a range of latitudes in the North Atlantic (Fig. 4). The time mean contribution to heat transport in the tropical North Atlantic is approximately 0.8 PW (Fig. 4a), which is weaker than observational (Johns et al., 2011) and model-based (e.g., Tréguier et al., 2012) estimates. The weaker Atlantic MHT in POP is likely associated with a weaker meridional overturning circulation of 12-13 Sv, which is towards the low end of model estimates (8–28 Sv, e.g., Danabasoglu et al., 2014). Despite the probable low bias in the strength of the overturning, the overturning contribution to time mean heat transport is much larger than non-overturning components at low latitudes (Fig. 4a), in agreement with earlier findings (Bryan, 1982; Johns et al., 2011). North of the Gulf Stream separation near 38°N, the overturning contribution steadily decreases while the mesoscale contribution increases, reaching a peak at 43°N. The time mean mesoscale contribution is larger than the contribution of large-scale (non-overturning) processes in the range 39°-45°N, while to the north and south large-scale processes contribute more, likely driven by the strong subpolar and subtropical gyre circulations respectively. The residual contribution is negligible at all latitudes in the North Atlantic, implying that short-timescale (<5 day) variability does not substantially impact our assessment of contributions to time mean MHT.

The flux components have also been filtered to consider their contribution to interannual and decadal (ID) variability, by removing the seasonal cycle and applying a low-pass filter with a half-power cutoff period of 14 months (Fig. 4b). While the higher-frequency (synoptic, intraseasonal, and seasonal) variability of MHT is substantial, we focus on ID variability in order to highlight the rectified impacts of mesoscale dynamics that may be of interest for climate studies. In the North Atlantic overall, the large-scale contribution to ID variability remains comparable to the mesoscale contribution, except at 40°–41°N (where the mesoscale is larger) and poleward of 44°N (where the large-scale is larger). The residual's contribution to ID variability is relatively small, if not as negligible as its contribution to time mean MHT. The similarity in the ID standard deviations of

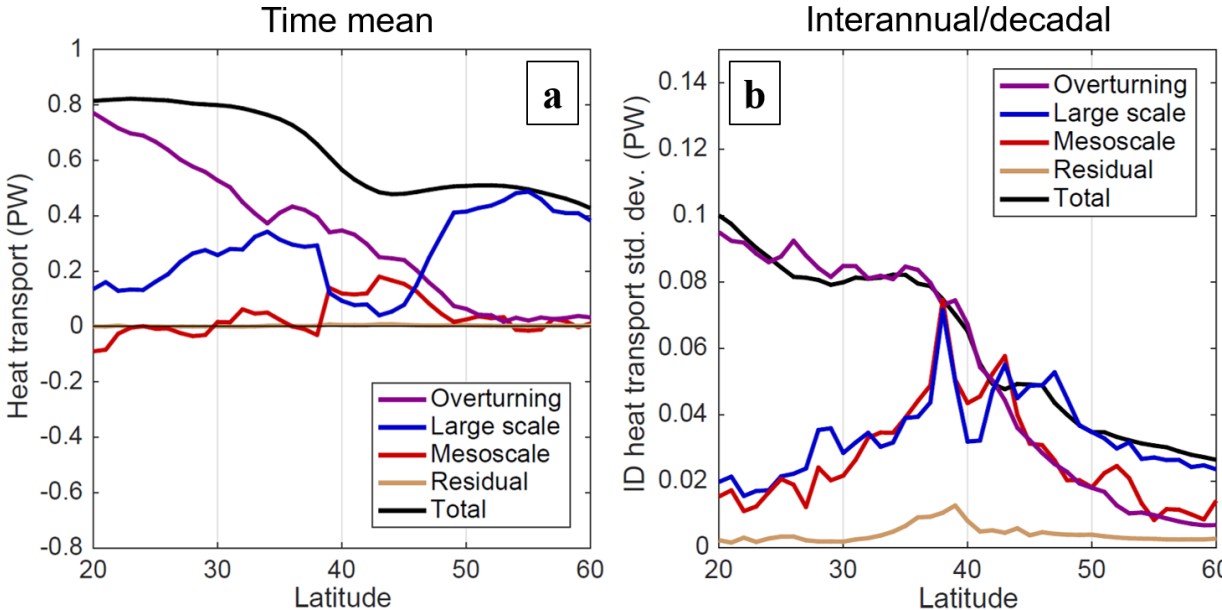

**Figure 4.** (a) POP time mean contributions to meridional heat transport (MHT) in the North Atlantic from overturning, large-scale, and mesoscale components, and the unexplained residual, as a function of latitude. (b) Standard deviation (on interannual/decadal timescales) of the components of MHT.

large-scale and mesoscale MHT variability at 30–38°N may also indicate compensation between the large-scale and mesoscale due to mesoscale feedbacks on the large-scale circulation (e.g., Hoskins et al., 1983; Waterman and Jayne, 2011) or large-scale

preconditioning of flow variability and temperature gradients where mesoscale dynamics are active. Therefore our method does not entirely disentangle the effects of the large-scale and mesoscale flow on temperature fluxes and transport. However, it provides a more precise diagnostic for the flux directly associated with mesoscale velocity and temperature anomalies; the spatial and temporal variability of these anomalies may then be studied in the context of variability in the background (large-scale) state.

In the rest of the analysis, we focus on 40°N to highlight how the spatial-scale decomposition can help diagnose the mechanisms of temperature flux variability across a transect. The 40°N latitude is an ideal location for this analysis, as the mesoscale contributions to time-mean and interannual/decadal (ID) MHT are both substantial, and higher than large-scale contributions at the same latitude (Fig. 4). Though the overturning contributions are still larger than the mesoscale at 40°N, we will disregard the overturning contributions in the remainder of this study to focus on the novel large-scale/mesoscale decomposition.

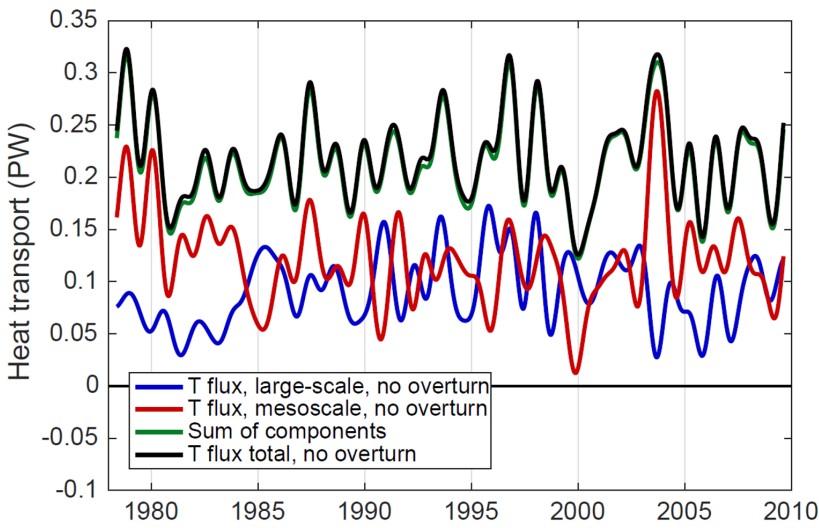

**Figure 5.** Basin-integrated contributions of large-scale and mesoscale $v$ and $T$ to the ID variability of heat transport across 40°N in the Atlantic. The zonal means of $v$ and $T$ are removed prior to the computations. The sum of the large-scale and mesoscale components (green) is compared to the total temperature flux minus the overturning contribution (black); differences between the two are due to high-frequency co-variances of $v$ and $T$ at timescales shorter than 5 days.

## 3.2 Time series of ID variability

With the overturning contribution removed, the time series of the large-scale and mesoscale temperature flux components at 40°N (Fig. 5) confirms that the ID variability of the MTF (red curve) is larger than the variability large-scale temperature flux (LTF, blue curve). The LTF variability is not negligible however, and some peaks in the total non-overturning temperature flux (black curve) such as in 1993 can be attributed to the LTF more than the MTF. At some times the LTF and MTF are anticorrelated and effectively cancel each other out (1990, 1995), and at other times both fluxes contribute substantially to a temperature flux peak (1979, 1996). Yet some of the highest peaks in the total flux (1980, 2003) are associated only with MTF variability, and the lowest total flux in the entire series (beginning of 2000) is also a result of low MTF. The sum of the two components (LTF and MTF) agrees very well with the total non-overturning temperature flux, implying a small residual from high-frequency (<5 days) $v$ and $T$ co-variability.

## 3.3 Cross-basin structure of flux components

We now turn our attention to the parts of the transect that contribute most to the time mean and ID variability of the large-scale and mesoscale temperature fluxes. The time-mean LTF is the result of three areas of contribution in the upper ocean (<300 meters), as well as two areas at depths of 400-1200 meters (Fig. 6a). The upper ocean contributions to time-mean LTF are mostly found near the western boundary, where a positive LTF closest to the boundary is partially compensated by negative

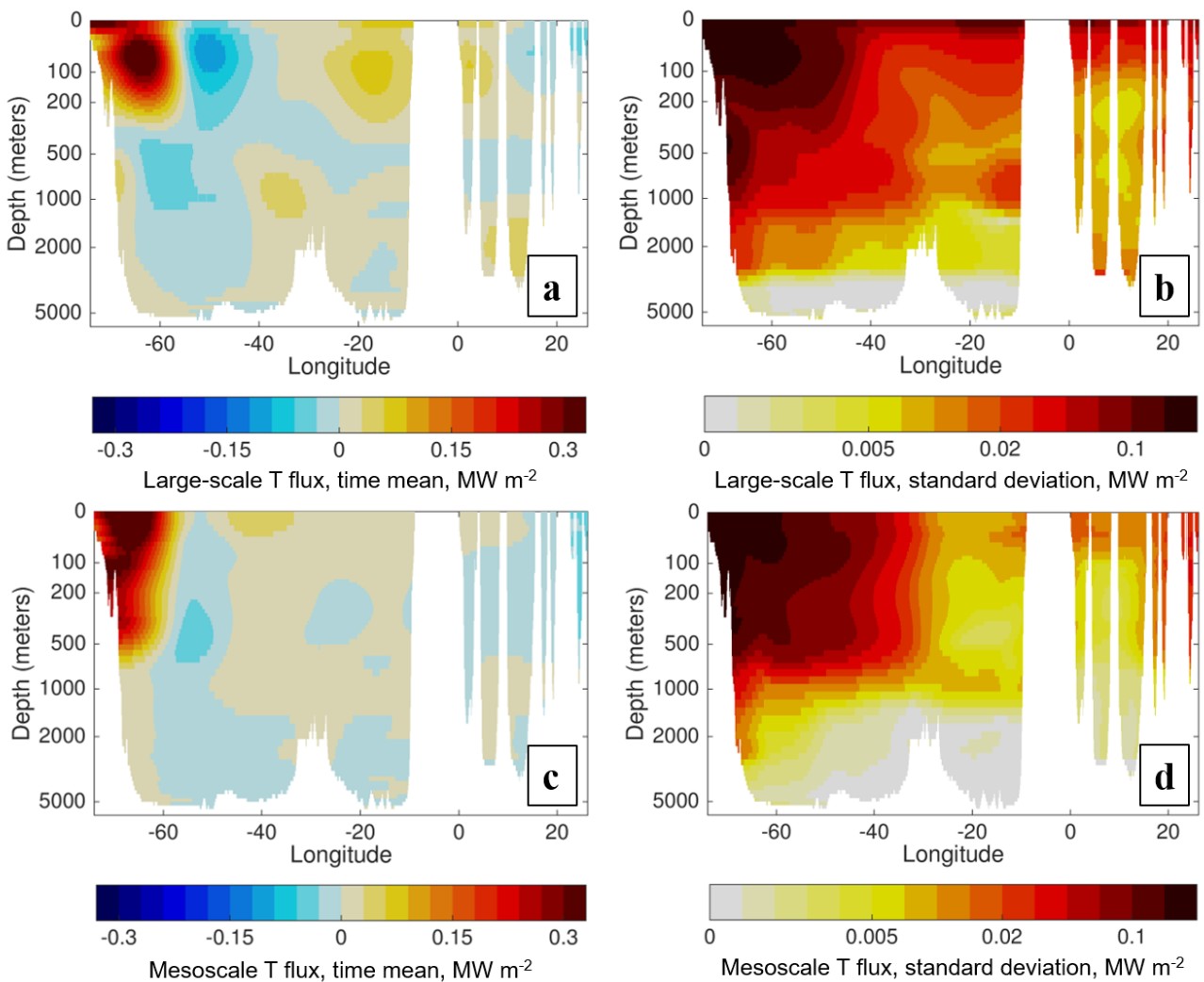

**Figure 6.** (a) Time mean and (b) ID standard deviation of the zonally-smoothed large-scale temperature flux. Zonal smoother is the low-pass filter in eq. (4) with $\lambda_0 = 20°$ longitude and $s = 2$. (c, d) Same as (a, b), but for the zonally-smoothed mesoscale temperature flux.

LTF further offshore near 50°W. A more modest positive LTF contribution is also located near the eastern boundary of the main ocean basin (20°-10°W). The 400–1200 meter contributions are between the western boundary and the Mid-Atlantic Ridge. These contributions of negative LTF at 65°-45°W and positive LTF near the Mid-Atlantic Ridge are more substantial than they may appear in Fig. 6a, since they span a larger depth range (note the logarithmic depth scale of the figure); indeed, most of the mid-ocean structure in the LTF as seen in Fig. 3l is associated with this deeper minimum and maximum. The LTF contribution

to ID variability is higher in the western part of the basin, especially near the western boundary where it is substantial to at least 3000 meters (Fig. 6b). The time mean MTF structure is simpler to interpret than the LTF, as it is vertically coherent and

mostly confined to the upper 700 meters (Fig. 6c). An area of positive MTF near the western boundary is stronger than the co-located positive LTF, the negative MTF at 60°-50°W is weaker than its LTF counterpart, and there is an additional modest area of positive MTF near 40°W. The ID variability of the MTF is confined almost entirely to the upper 1500 meters, and west of the Mid-Atlantic Ridge (Fig. 6d).

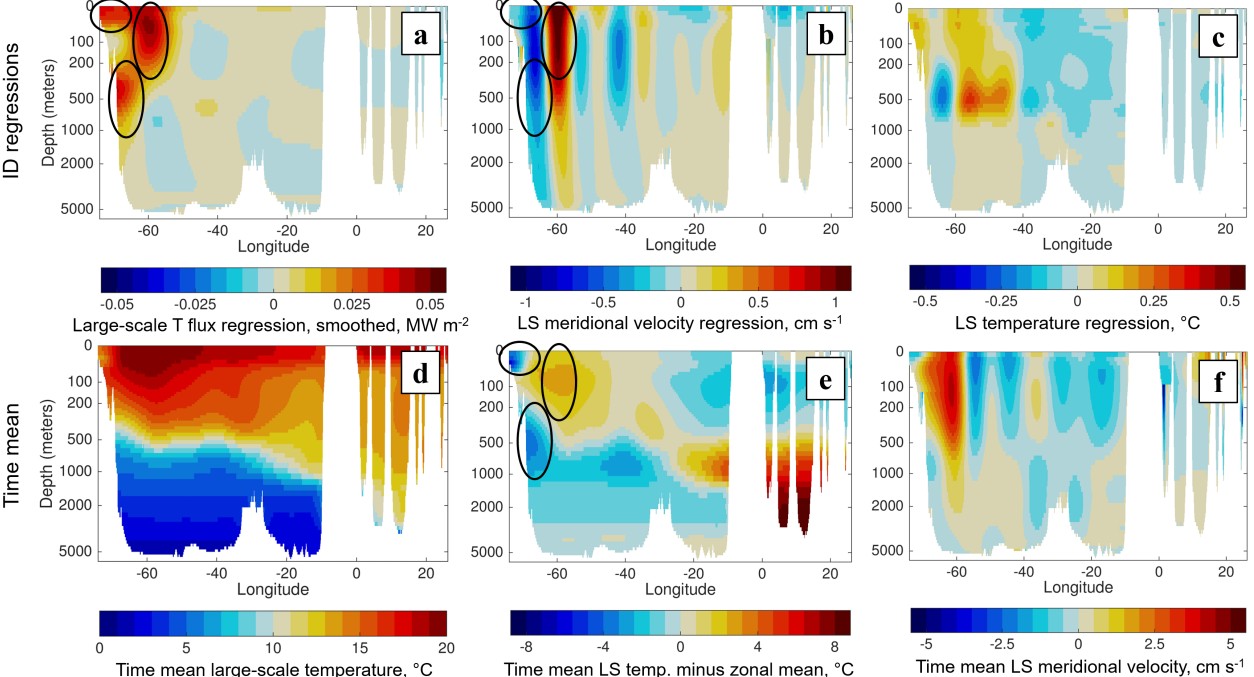

**Figure 7.** Components of the large-scale temperature flux variability. (a) Regression of $+1\sigma$ of the basin-integrated large-scale (LS) temperature flux onto the local zonally-smoothed LS temperature flux, and onto unsmoothed (b) LS meridional velocity and (c) LS temperature. (d-f) Time mean, unsmoothed structure of LS temperature (d) with zonal mean and (e) without zonal mean, and (f) time mean LS velocity structure. Ellipses indicate areas of contribution to the LTF from time-variable LS meridional velocity and time-mean LS temperature $v_L' \overline{T_L}$.

The ID variability of the LTF can be explained in terms of the structure of its large-scale velocity and temperature constituents (Fig. 7). Using linear regression, we assess the local contributions of the LTF associated with a $+1\sigma$ (1 standard deviation above the mean) basin-integrated LTF (Fig. 7a). In Figure 7a the LTF is zonally-smoothed using the same filter as in Figure 6, to facilitate interpretation and reduce the appearance of zonal LTF variability at wavenumbers up to twice that of the individual velocity and temperature constituents. The regression of the basin-integrated onto local LTF reveals most of the variability is found west of 50°W. Of the three main areas of contribution (positive regression values, indicated by solid ellipses), one extends to nearly 1000 meters depth, with water temperatures of 5°-12°C (Fig. 7d). In this depth range the coldest waters are found at the western boundary, and the LTF contribution increases when the flow is southward (Fig. 7b), as the boundary temperature is

cold relative to the zonal mean (Fig. 7e). The other areas of contribution to LTF variability are near the surface, most notably the maximum near 60°W associated with the near-surface maximum in temperature (Fig. 7e), where northward flow increases the LTF (Fig. 7b). A third smaller area near the coast is associated with the advection of relatively cold surface waters, and there southward advection increases the LTF. All of these contributions are associated with velocity variability advecting time-mean temperature structure ($v'_L \overline{T_L}$). The time mean velocity field does also advect temperature variability ($\overline{v_L} T'_L$), but the effects of this term (Fig. 7c,f) are too small to be represented in any of the maxima in Figure 7a.

The structure of the zonally-smoothed MTF variability on ID timescales (Fig. 8a) has some similarity to the structure of the time-mean MTF; however, the interior region (50°-35°W) which had a modest positive time-mean MTF becomes much more important for ID variability. The focus of this interior MTF is also deeper than the boundary MTF, near 500 meters depth where the thermocline shoals (Fig. 7d). The western boundary remains significant for MTF variability; the middle region (60°-50°W) has a much smaller effect on ID variability compared to the western boundary and interior MTF contributions (Fig. 8a). Given the predominance of transient, propagating features in the oceanic mesoscale (e.g., eddies), much ID MTF variability at this latitude is not associated with the time-variable advection of time mean $\overline{T_M}$, but with the rectification of intraseasonal $v'_M$ and $T'_M$ variability onto ID timescales. Therefore, the regression method used to explain the sources of LTF variability in Fig. 7 is insufficient to explain sources of the MTF, and different diagnostics are needed.

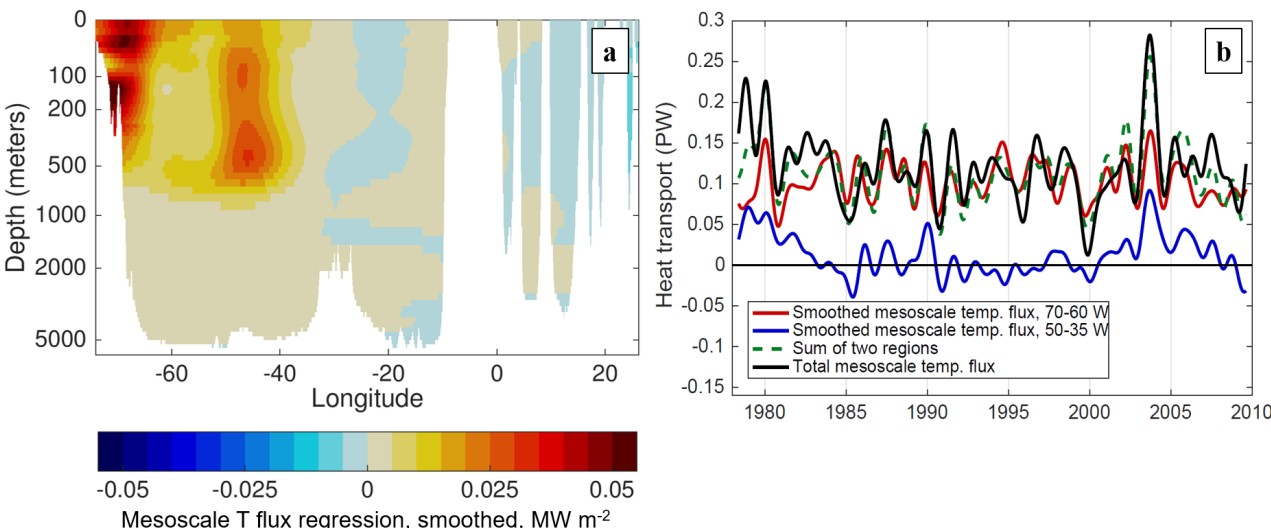

**Figure 8.** (a) Regression of +1$\sigma$ of the basin-integrated mesoscale temperature flux onto the local zonally-smoothed mesoscale temperature flux, indicating the distribution of the temperature flux contributions along the transect. (b) Time series comparison of the zonally-smoothed mesoscale temperature flux integrated in two regions (70–60° W and 50–35° W) and the sum of the contributions in these regions with the total basin-integrated mesoscale temperature flux.

Figure 8a implies that two regions (a boundary and interior region) should account for most MTF variability on ID timescales. Time series of the MTF in each of these regions (Fig. 8b) confirm this while also illustrating differences in the timescales of variability present in each region. The variability of MTF in the boundary region (70°-60°W) predominantly occurs on higher-frequency interannual timescales with peaks every 1–4 years, and many of these peaks are aligned with the basin-integrated MTF. The interior region (50°-35°W) has some interannual variability, but there also appears to be an underlying decadal signal, and elevated interior MTF peaks in 1980 and 2003 align with some of the highest values of basin-integrated MTF during this period.

## 4 Drivers of MTF variability

### 4.1 EKE variability and MTF

The eddy kinetic energy or EKE (defined as half the variance of the velocity vector with the time mean removed) is generally considered to be an indicator of the level of mesoscale activity in a given location and time; hence we expect that EKE might influence MTF variability. This relationship can be expressed (e.g., Holloway, 1986; Stammer, 1998) as

$$v_M T_M = \kappa \frac{\partial T}{\partial y} \propto \sqrt{v'^2} L_{\mathrm{mix}} \left( -\frac{\partial T}{\partial y} \right) \tag{8}$$

where $L_{\mathrm{mix}}$ is a mixing length parameter that is related to the width of the local frontal zone associated with the meridional temperature gradient (Green, 1970). Assuming fairly isotropic velocity variability

$$v_M T_M \propto \sqrt{\mathrm{EKE}} L_{\mathrm{mix}} \left( -\frac{\partial T}{\partial y} \right) \tag{9}$$

It would be convenient if EKE (specifically surface EKE) was the primary influence on MTF variability, since this can be diagnosed from satellite altimeters and would allow direct observational estimates of MTF variability. However, at 40°N surface EKE variability is not a very reliable indicator of time variations in the MTF (Fig. 9). High MTF generally occurs at times of high transect-averaged EKE, and significant MTF peaks are all associated with surface EKE peaks. However, many of the highest values of surface EKE (1985-86, 1992, 2004, 2008) do not seem to drive an increased MTF; therefore, elevated EKE seems to be a necessary but not sufficient condition for high MTF along this transect. Local correlations of surface EKE and MTF in the eddy-active western part of the basin (not shown) are at best marginally significant at 95% confidence levels, further suggesting that high levels of mesoscale energy do not imply elevated MTF.

### 4.2 Constituents of MTF variability

To focus on what conditions permit high MTF across 40°N, we consider three episodes of high MTF, in 1980, 1996, and 2003 (Fig. 9). The spatial structure of MTF variations indicates that all of these peaks are associated with higher than usual MTF

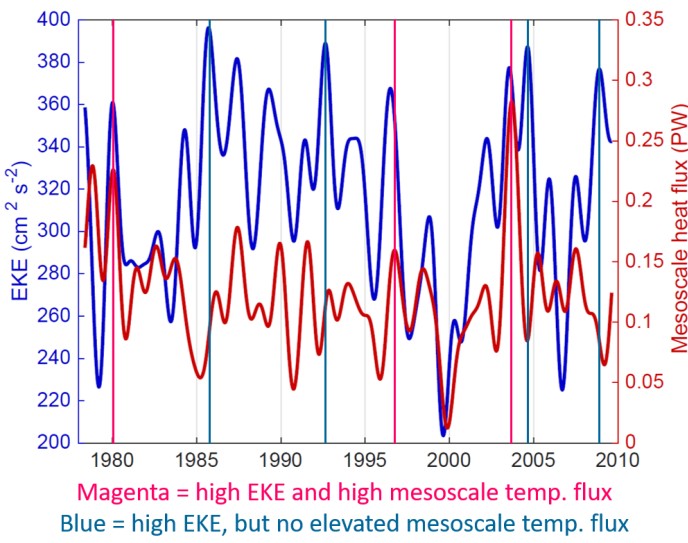

Magenta = high EKE and high mesoscale temp. flux
Blue = high EKE, but no elevated mesoscale temp. flux

**Figure 9.** Comparison of the basin-integrated mesoscale temperature flux (red) with the zonally-averaged surface eddy kinetic energy (EKE; blue) along 40°N. The vertical lines indicate times of (magenta) high EKE and elevated MTF, vs. (blue) high EKE but without elevated MTF.

values in more than one region (Fig. 10a,b). In 1980 and 2003 (the two highest peaks) the elevated MTF contributions clearly originate in the boundary and interior regions. In 1996, a positive MTF anomaly in the boundary region is supplemented by a broad (but not particularly strong) positive anomaly that extends eastward to ∼45°W (Fig. 10b), which corresponds to an abatement of the usually negative MTF in this region (Fig. 10a).

Typically in the interior of the ocean, the dominant contribution to the MTF is associated with the $v_M T_M$ term in eq. (3). Hence in addition to the proportionality relationship in eq. (9), the MTF can be decomposed in terms of the contribution of the amplitude and co-variability of $v_M$ and $T_M$

$$v_M T_M = \sigma_{v_M} \sigma_{T_M} R \tag{10}$$

where $\sigma_{v_M}$ and $\sigma_{T_M}$ are the standard deviations of $v_M$ and $T_M$ and $R$ is the correlation coefficient of $v_M$ and $T_M$; the standard
deviations and correlations are computed in windows that are localized in space and time to study spatiotemporal variability. In relation to eq. (9), EKE is most likely to influence $\sigma_{v_M}$, while the meridional temperature gradient $\partial T/\partial y$ will likely influence $\sigma_{T_M}$ and potentially $R$. By computing the constituents on the right-hand side of eq. (10) in moving windows spanning 10° longitude and 1 year ranges (in the upper 1000 meters), the contributions of each constituent to MTF variability are evident (Fig. 11a-c). The basin-integrated MTF is the result almost entirely of fluxes west of the Mid-Atlantic Ridge (∼30°W) because
$\sigma_{v_M}$ and $\sigma_{T_M}$ are much larger in the western part of the basin than in the east (Fig. 11a-b). However, $R$ is responsible for the time-mean structure of the MTF within the western part of the basin; in fact, $R$ is typically negative between 60°-50°W despite

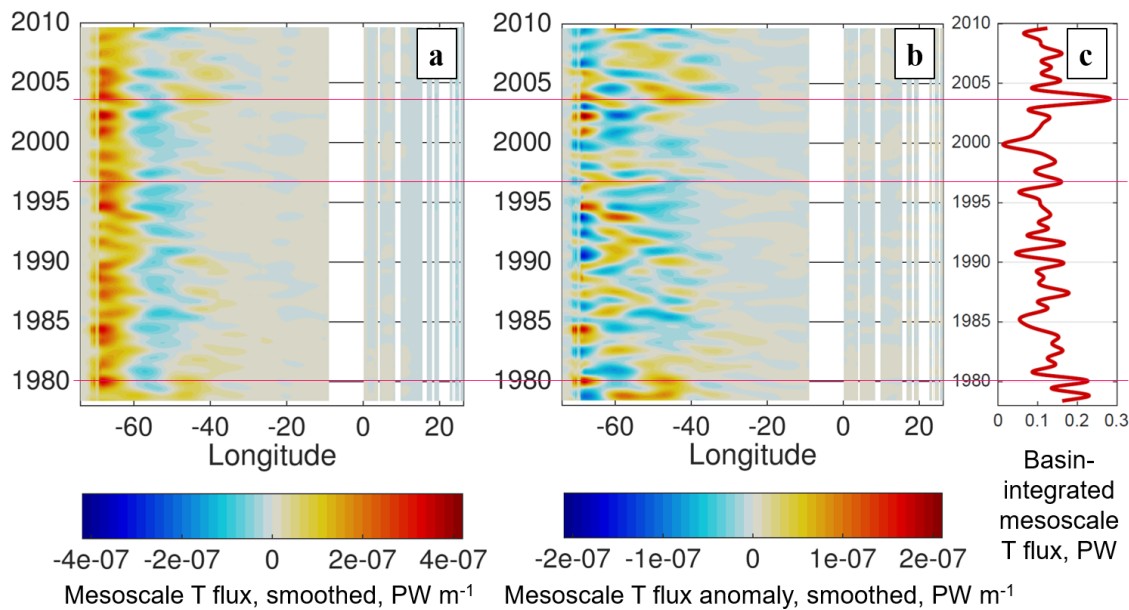

**Figure 10.** (a) Hovmöller diagrams of the zonally-smoothed mesoscale temperature flux at 40°N. (b) Same as (a) with the time mean removed. (c) Time series of the full basin-integrated mesoscale temperature flux. Magenta lines indicate the times of highest mesoscale contribution to meridional heat transport.

the fact that the large-scale meridional temperature gradient has the same sign here as elsewhere (Fig. 11d). This would imply an upgradient flux of temperature at 60°-50°W, though it can not be determined solely from this zonal transect whether this apparent upgradient diffusivity is associated with a rotational component of the temperature flux (e.g., Marshall and Shutts, 345   1981).

When the time means are removed from the constituents in eq. (10), the sources of MTF variability can be attributed more clearly (Fig. 12). In 1980, high MTF is driven by a strongly positive $R$ anomaly, coincident with a steeper-than-usual temperature gradient in the interior region (Fig. 12c,d). By contrast, high MTF in 1996 is supported by slight positive anomalies in $\sigma_{v_M}$ and $R$ (and possibly $\sigma_{T_M}$), while in 2003 has more robust positive anomalies in all of these contributions. The anomalies 350   also suggest a difference in behavior between the boundary and interior regions: the MTF is more responsive to $\sigma_{v_M}$ (and likely EKE) variations west of 50°W where time-mean $\sigma_{v_M}$ is larger, while in the interior an increase in $\sigma_{T_M}$ and/or $R$ is necessary to increase MTF. Lastly, all three peak events (Fig. 10) are associated with an anomalously steep meridional temperature gradient in their source regions (Fig. 12d). Just as importantly, there was no robust positive (weaker) gradient anomaly during these three events, in contrast with 1992-95 when weaker gradients in the interior region may have contributed to negative (weak) 355   anomalies in all three constituents.

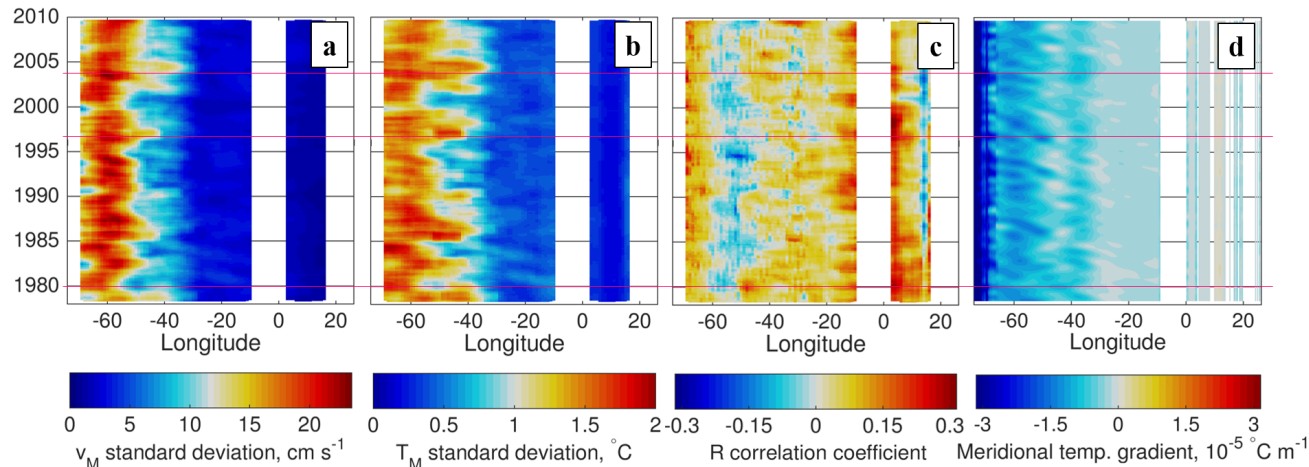

**Figure 11.** (a-c) Breakdown of mesoscale temperature flux according to the decomposition $v_M T_M = \sigma_{v_M} \sigma_{T_M} R$, where $\sigma_{v_M}$ and $\sigma_{T_M}$ are the local standard deviations of mesoscale $v_M$ and $T_M$, and $R$ is the local correlation coefficient between them. Standard deviations and correlations are computed for the upper 1000 m of the ocean, in moving windows spanning $10°$ longitude and 1 year. (d) Meridional temperature gradient, determined using a regression fit between latitudes $38°$ N and $42°$ N and zonally and temporally low-pass filtered with $10°$ and 14 month thresholds respectively.

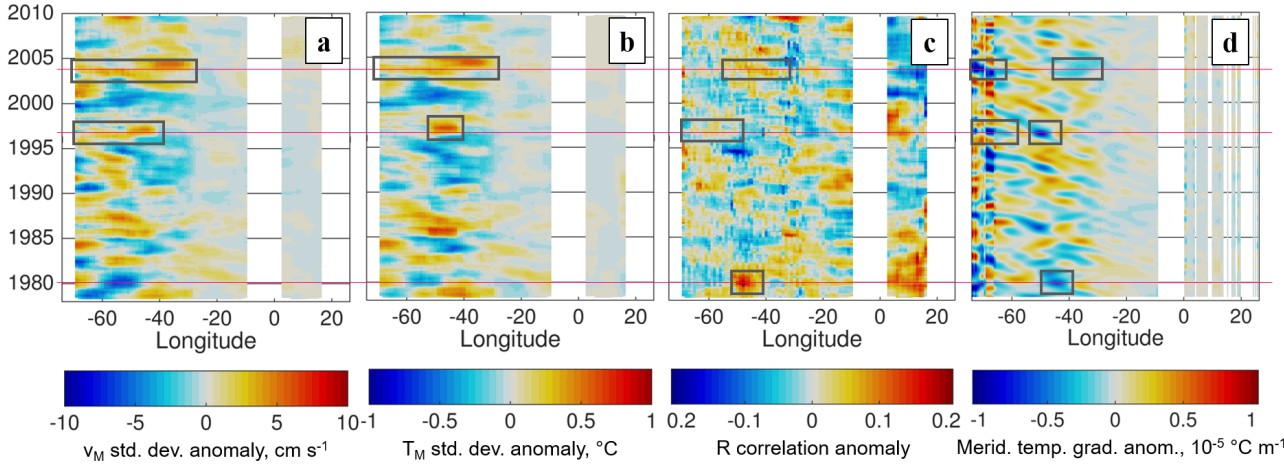

**Figure 12.** Same as Fig. 11, but with the time means removed. Anomalies associated with high MTF events in 1980, 1996, and 2003 are indicated by gray boxes.

## 4.3  Large-scale temperature gradient contributions

Having identified the influence of the meridional temperature gradient on MTF variability, we consider the role of the large-scale current and temperature in generating anomalous meridional temperature gradients. The path of the Gulf Stream extension sets the location of strong temperature fronts in the western Atlantic at this latitude; notably, its path during high MTF events

tends to be further north than usual at ∼66°W (Fig. 13a). This location is important because in the POP simulation it is where the mean path of the Gulf Stream first approaches 40°N. When there is a northward shift in the path at this region, the Gulf Stream's crossing of 40°N can happen hundreds of kilometers west of the usual location, and the meridional temperature gradient is steeper west of 65°W (Fig. 11d). Large-scale temperature anomalies in the upper ocean also show a difference in the mechanism for meridional temperature gradient anomalies at the boundary vs. the interior (Fig. 13b-d). When the boundary

region contributes significantly to high MTF, the steeper temperature gradient is associated with a positive temperature anomaly south of 40°N. When the interior region contributes to high MTF, the steeper gradient is associated with a negative temperature anomaly north of 40°N. Since the mean path of the Gulf Stream is just south (north) of 40°N in the boundary (interior) region, each of these temperature anomaly patterns would nudge the path of the Gulf Stream closer to 40°N, intensifying the temperature gradient across the transect.

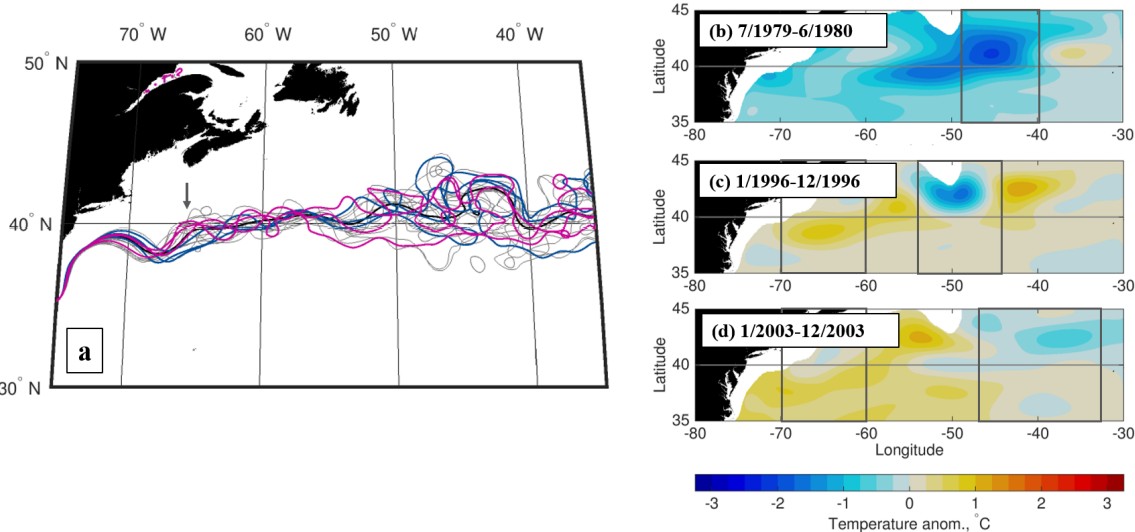

**Figure 13.** (a) Spaghetti plot showing annual averages of the -20 cm contour path (approximate Gulf Stream path) in POP; magenta lines indicate the path during times of high EKE and MTF, blue lines indicate times of high EKE but low MTF, and gray lines are randomly distributed 1-year average paths during 1978–2009. The arrow indicates a path anomaly associated with the high MTF events. (b-d) Large-scale temperature anomalies, averaged 0–1000 meters, during the periods of high mesoscale temperature flux. The gray boxes indicate the region(s) in which negative meridional temperature gradients contribute the most to the mesoscale temperature flux.

The importance of the meridional temperature gradients in both the boundary and interior regions is emphasized when comparing the time series of the gradients to the MTF in each region (Fig. 14). In particular, the higher-frequency interannual variability in MTF in the boundary region is closely associated with variations in the meridional temperature gradient (Fig. 14a). In the interior region, decadal variability is not as pronounced in the meridional temperature gradient as it is in the MTF; there are a number of times when the meridional temperature gradient is steeper than the average, without a major effect on

MTF (Fig. 14b). However, two of the most significant maxima in meridional temperature gradient (1979-80 and 2003-04) are associated with elevated MTF, so steeper gradients at least make significant positive MTF events more likely.

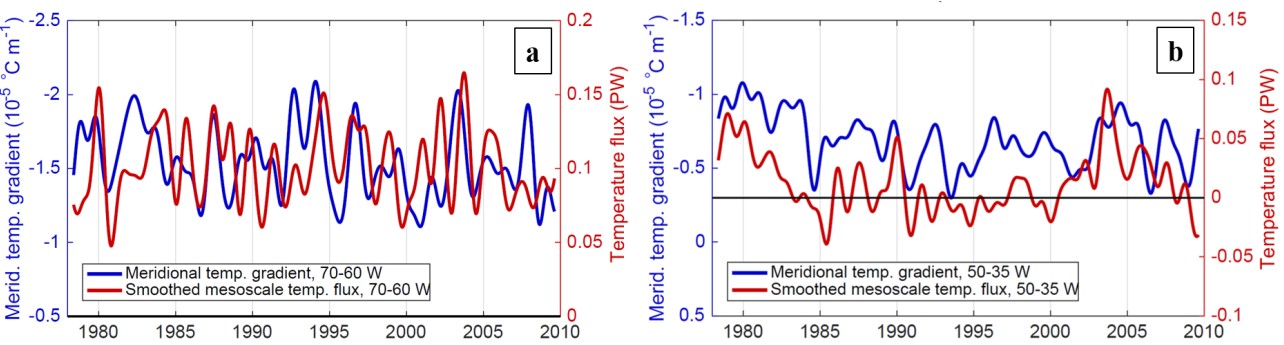

**Figure 14.** (a) Time series comparison of 0–1000 m meridional temperature gradient averaged along the 40° N transect between 70° W and 60° W (boundary region), with the zonally-smoothed mesoscale temperature flux integrated in the same region. The y-axis of the meridional temperature gradient is inverted to align its orientation with the MTF. (b) Same as (a), but in the interior region (50°-35° W).

## 5   Discussion

### 5.1   Novel aspects of this study

    A key novel aspect of this study is the spatial decomposition method to separate the mesoscale and large-scale contributions to

MHT, in contrast to the use of temporal co-variability of velocity and temperature in many previous studies. As a diagnostic tool, the spatial-scale decomposition of MHT has important advantages over the more common approach of separating time mean and time deviation (often called "eddy") fluxes. The mechanisms that drive large-scale processes such as gyres (wind forcing, boundary/topographic constraints) are often distinct from those driving mesoscale processes (e.g., baroclinic and barotropic instability, nonlinear momentum/vorticity advection); quantifying the spatial scales of MHT contributions and their

distribution is a first step towards understanding the processes that contribute to MHT. Moreover, rather than being defined relative to a time mean over an arbitrary time period (e.g., the post-spinup time span of the model simulation), the mesoscale MHT is defined as the deviation from a regional background state at each time. Since the integrated volume flux associated with

the mesoscale velocity is small across zonal scales $\gg 10°$ longitude, the MTF smoothed over large zonal scales approximates a heat transport regionally and at each time.

Exploiting the advantages of the spatial decomposition method, this study outlines an approach for diagnosing the geographical origins of LTF and MTF variability. Large-scale temperature fluxes can generally be explained in terms of large-scale velocity and temperature components directly, with at least one time mean component involved (Figure 7). Mesoscale processes such as transient eddies often produce rectified fluxes at spatial (and temporal) scales that are much larger (longer) than the original velocity and temperature anomalies. However, the MTF contributions can be traced to specific locations (Fig. 6b,8), and supplemented by an analysis of the conditions that influence the MTF and instability in those regions (Fig. 9–14).

In our analysis of MTF variability at $40°$ N, we found that the influence of the meridional temperature gradient is modulated by the velocity-temperature correlation, whose importance is emphasized by Fig. 11–12. In addition to its role in driving MTF temporal variability, the correlation explains the negative time-mean temperature flux at $60°$-$50°$ W, suggesting that there is an upgradient flux across the meridional temperature gradient (Fig. 11c,d). Other studies of mesoscale eddy characteristics and heat fluxes (e.g., Hausmann and Czaja, 2012; Gaube et al., 2015; Frenger et al., 2015; Tréguier et al., 2017) have considered the displacement of temperature relative to velocity/pressure anomalies in eddies. Yet this work illustrates the impact of even subtle changes (of order 0.1) in the velocity-temperature correlation coefficient; hence more comprehensive studies of the velocity-temperature correlation, its dependence on the structure of nearby fronts and relationship to existing theories of diffusivity are needed.

## 5.2   Comparison of MTF with previous formulations of the eddy flux

It is helpful to compare how the mesoscale MHT we estimated compares to those based on previous methods, i.e., the co-variability of velocity and temperature (1) for all time scales ("time-varying"; e.g., Jayne and Marotzke, 2002; Tréguier et al., 2017), (2) for deviations from 3 month-averages only ("high frequency"; Volkov et al., 2008), and (3) the baroclinic "eddy" contribution, which Hall and Bryden (1982) defined by removing first the depth-averaged (barotropic) and then the zonal average $v$ and $T$. Because of the differences in models and observations used in these studies, a direct comparison of mesoscale MHT in our analysis with "eddy" heat transports from these previous studies is not suitable. We therefore compare our mesoscale MHT with the "eddy" MHT based on previous methods using the same POP model output (Fig. 15). The time-varying $v'T'$ term happens to be a decent approximation of the mesoscale contribution to the time mean MHT in the $39°$–$43°$N latitude range (Fig. 15a), corresponding to the region of highest MTF just north of the Gulf Stream separation. To the south and north of this active mesoscale region, all of the "eddy" formulations have much lower time-mean values, with the exception of the baroclinic eddy term which peaks as high as 0.35 PW at $36°$N. However, the definition of the baroclinic eddy flux includes large-scale gyre flows that have a baroclinic component, and the baroclinic eddy contribution is generally comparable to or smaller than the large-scale contribution to time-mean MHT (Figure 4a). The negative and positive mesoscale contributions to MHT at $28°$N and $34°$N respectively (Figure 3) can be seen in Figure 15a, and since these contributions are not found in the time-varying MHT we can infer that this mesoscale MHT is associated with the time-mean (stationary) structure of the Gulf Stream and its tight southward recirculation.

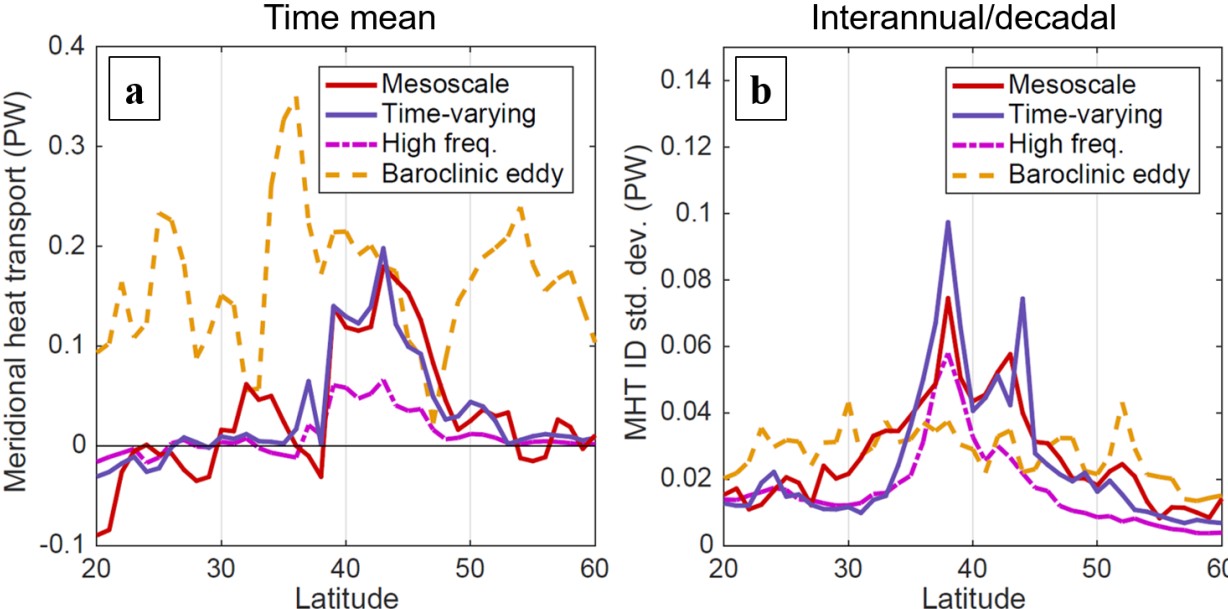

**Figure 15.** (a) POP time mean contributions to meridional heat transport (MHT) in the North Atlantic from "eddy" formulations computed four ways: the mesoscale component computed as the bracketed portion of eq. (3), the time-varying component $v'T'$ (e.g., Jayne and Marotzke, 2002), the high frequency component using only $v'$ and $T'$ on timescales shorter than 3 months (Volkov et al., 2008), and the baroclinic eddy component as defined in Hall and Bryden (1982). (b) MHT standard deviation (on ID timescales) of the four "eddy" formulations.

Of the contributions to time-mean MHT, intraseasonal (high) frequencies account for about 30–40% of the total time-varying contribution in the active eddy region north of the Gulf Stream separation (Fig. 15a). While mesoscale eddies are typically associated with intraseasonal frequencies, in the strong eastward flow of the Gulf Stream the westward propagation
of eddies and meanders is slowed and even in some cases reversed, resulting in more low-frequency (and stationary) mesoscale variability driving MHT. Regarding ID variability as a function of latitude, the time-varying flux has steep spikes in variability near the edges of the active mesoscale range at 38° and 44°N; the mesoscale contribution is again similar to the time-varying in the 39°–43°N range but the mesoscale has more variability south of the Gulf Stream separation (Fig. 15b). The baroclinic eddy component has consistent variability across latitudes, generally lower than that of the time-varying and mesoscale fluxes
at 35°–45°N and higher outside of this range.

## 6   Conclusions

In this study a new decomposition method has been used to distinguish the contributions of mesoscale vs. larger-scale processes to meridional heat transport in the North Atlantic by using spatial scales (rather than temporal deviations) of velocity and

temperature. This analysis technique can be applied to eddy-permitting ocean and coupled GCMs to better quantify the tem-

perature flux produced by mesoscale ocean processes in the model. Applying this spatial-scale decomposition method in the North Atlantic, a substantial mesoscale contribution to time-mean, non-overturning MHT was found in the $39°–45°$ N latitude range that exceeded the large-scale non-overturning contribution, while somewhat less than the overturning contribution (Fig. 4a). North of the Gulf Stream separation the mesoscale contribution is associated with time-variable fluxes and so it is similar to the contribution of the traditional "eddy" temperature flux (Fig. 15a). However, south of the Gulf Stream separation there is

a mesoscale contribution to time mean MHT that is associated with stationary mesoscale structures and is not included in the time-varying $v'T'$ term. Since the mixing effects of mesoscale processes apply to both stationary and time-variable processes, the mesoscale temperature flux is a more meaningful estimate of the mesoscale contribution to MHT that should be represented by eddy parameterizations in non-eddying models (e.g., Gent and McWilliams, 1990).

This study has also considered the relationship of mesoscale temperature flux variability to variations of indicators such as

the EKE and meridional temperature gradient. The first unexpected result is that eddy kinetic energy (or at least surface EKE) is not a reliable indicator of MTF variability, with many instances of zonally-averaged surface EKE not being associated with an elevation in the MTF (Fig. 9). It is not surprising that meridional temperature gradients influence MTF variability, given that cross-frontal gradients are a part of classic theories of diffusivity and lateral mixing dating back at least to Taylor (1915). Yet the low magnitudes of the velocity-temperature correlation $R$ imply that even small changes in $R$ can have a large impact

proportionally on the MTF. Hence an improved understanding of velocity and temperature structure within mesoscale features is necessary to inform accurate representations of meridional temperature fluxes in models.

*Data availability.* The POP model output used in this study is stored on NCAR's High Performance Storage System (HPSS); the full model output in 5-day averages is available with a user account (through https://www2.cisl.ucar.edu) by logging into cheyenne.ucar.edu and accessing the following path on HPSS: /home/bryan/johnsonb/g.e01.GIAF.T62_t12.003/ocn/hist/. Source code to run the POP2 model is

available at http://www.cesm.ucar.edu/models/cesm1.0/pop2/. The CMEMS surface dynamic topography data used to produce the analysis in Figure 1 are available from http://marine.copernicus.eu/services-portfolio/access-to-products/ by searching for the Product ID SEALEVEL_GLO_PHY_L4_REP_OBSERVATIONS_008_047.

*Author contributions.* Primary author Andrew Delman wrote the code, carried out the analysis presented, and drafted the manuscript. Tong Lee supervised the project, providing input into the direction of the research and edits to the manuscript.

*Competing interests.* There are no competing interests present in the publication of this paper.

*Acknowledgements.* The research was carried out at the Jet Propulsion Laboratory, California Institute of Technology, under a contract with the National Aeronautics and Space Administration (80NM0018D004) with the support of NASA Physical Oceanography. The authors would like to acknowledge Benjamin Johnson who ran the POP model configuration and made the output available, as well as Frank Bryan and Steve Yeager at the National Center for Atmospheric Research for their correspondence regarding model biases in the North Atlantic. The authors also acknowledge the helpful insights of two anonymous reviewers whose comments prompted substantial improvements to this manuscript.

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
