# Peer review of "A new method to assess mesoscale contributions to meridional heat transport in the North Atlantic Ocean"

_Ocean Science, 2020_

## Referee Comment (RC1) · Anonymous Referee #1 · 30 Apr 2020

Review of "A new method to assess mesoscale contributions to meridional heat transport in the North Atlantic Ocean" by Delman and Lee

This manuscript presents interesting results regarding the origin of meridional heat transport fluctuations in the North Atlantic. It is well written and to the point. I recommend publication with minor revisions.

Major comments: 1.The decomposition used by the authors, although mathematically correct, might not represent the effects of the "mesoscale" better than the other meth-ods cited in the introduction. What I have in mind is not so much spatial vs temporal scales but the fact that "mesoscale" fluctuations drive mean circulations which themselves carry heat. These two effects tend to compensate each other so the real effect of the "mesoscale" remain unclear (see for example the Transformed Eulerian Mean framework and its use in discussing tracer transports in the atmosphere –usually some dissipation or diabatic effects are needed to make the compensation imperfect). The new decomposition, although clearly of interest, does not shed light on this and this should be mentioned somewhere in the text. Another way to state this is that it is not clear if the heat transport captured by the author really is a heat transport: a warm blob could be advected poleward at depth (i.e., shielded from air-sea interactions and diabatic effects) for a while and then returned equatorward, still below the mixed layer, without creating a net heat transport in the mean although, in a timeseries, it would show up as an enhanced, then a decreased heat transport. 2.It might be more natural to analyse the heat transport along a mean streamline rather than across 40N, especially considering the model grid isn't latitude-longitude. This could remove some of the difficulties associated with steady meanders in the model Gulf Stream with scales comparable to those of the "mesoscale" and simplify the physical interpretation of the results (e.g., the cancellations between poleward and equatorward heat transport in Fig. 5a). I understand this isn't trivial to do but if the authors can do it within a reasonable amount of time it would add to the quality of the analysis.

Minor comments: 1.The length of the integration should be mentioned in the text (I only had a hint of it by looking at the period covered by the timeseries displayed –I couldn't see this information in the model description). 2.line 290: "upgradient heat transport" needs to be stated with caution. You might need to remove the rotational component of the heat transport vector first. 3.Abstract, line 5 "…that are more fundamental to the physics of ocean eddies". I am not sure the spatial scales are more fundamental than the temporal scales so maybe this could be removed.

---

## Referee Comment (RC2) · Anonymous Referee #2 · 18 May 2020

This paper is well written and I recommend publication after some revision/addition.

The paper points to its novel aspect a being a new way to do the separation of the eddy vs. large-scale contributions, and primarily as a different view than the Hall and Bryden (1982) separation of the baroclinic component of the heat transport. As such, I think the authors should directly compare some of the computations and maps to the Hall and Bryden method.

For example, in Figure 3, would the HB82 eddy term look different? Same in Figure 4, etc. Jayne and Marotzke (Rev. of Geophys. 2001) did some comparisons of the HB82 decomposition vs. the other time-varying heat transport terms.

[Figure]

Figure 5 seems to indicate that the separation between their large-scale vs. mesoscale is not very great. That is the spatial filter they used doesn't seem to really separate the spatial scales well, and the spectra in Figure show there isn't really a strong scale separation, especially in the temperature. It should be commented on.

———————————————————

---

## Author Comment (AC1) · 17 Jun 2020

Note: The original reviewer comments are indicated in italic font below, with the author responses and manuscript edits following in non-italic font.

This manuscript presents interesting results regarding the origin of meridional heat transport fluctuations in the North Atlantic. It is well written and to the point. I recommend publication with minor revisions.

Thanks to the reviewer for the very thoughtful and helpful comments on this manuscript. We have taken the suggestions into account as described below.

1. The decomposition used by the authors, although mathematically correct, might not represent the effects of the "mesoscale" better than the other methods cited in the introduction. What I have in mind is not so much spatial vs temporal scales but the fact that "mesoscale" fluctuations drive mean circulations which themselves carry heat. These two effects tend to compensate each other so the real effect of the "mesoscale" remain unclear (see for example the Transformed Eulerian Mean framework and its use in discussing tracer transports in the atmosphere –usually some dissipation or diabatic effects are needed to make the compensation imperfect). The new decomposition, although clearly of interest, does not shed light on this and this should be mentioned somewhere in the text.

The reviewer is correct that there may be compensation between the large-scale and mesoscale circulations. To better refine our definition of the mesoscale circulation, we have made some improvements to the large-scale/mesoscale decomposition method in coastal/boundary regions (as described in Section 2.2.3, lines 174-198) in order to ensure that the volume transport of the mesoscale circulation is negligible over long distances (Figure 3 in the new manuscript). However, our method only assesses the "direct" effect of mesoscale velocity and temperature anomalies on the meridional heat transport, just as the time-varying "eddy" heat flux assesses only the direct effect of the temporal co-variability of velocity and temperature anomalies. Neither the spatial nor the temporal decomposition methods fully separate the effect of mesoscale variability on the large-scale mean circulation, though the spatial decomposition of v and T could be a helpful diagnostic to understand the relationship between large-scale and mesoscale variability.

To clarify this point, the text has been revised as follows: "The similarity in the ID standard deviations of large-scale and mesoscale MHT variability at 30-38°N may also indicate compensation between the large-scale and mesoscale due to mesoscale feed-backs on the large-scale circulation (e.g., Hoskins et al. 1983; Waterman and Jayne 2011) or large-scale preconditioning of flow variability and temperature gradients where
mesoscale dynamics are active. Therefore our method does not entirely disentangle the effects of the large-scale and mesoscale flow on temperature fluxes and transport. However, it provides a more precise diagnostic for the flux directly associated with mesoscale velocity and temperature anomalies; the spatial and temporal variability of these anomalies may then be studied in the context of variability in the background (large-scale) state." (lines 237-244)

Another way to state this is that it is not clear if the heat transport captured by the author really is a heat transport: a warm blob could be advected poleward at depth (i.e., shielded from air-sea interactions and diabatic effects) for a while and then returned equatorward, still below the mixed layer, without creating a net heat transport in the mean although, in a timeseries, it would show up as an enhanced, then a decreased heat transport.

The reviewer seems to be referring to meridional heat transport (MHT) in a narrow sense, namely, the meridional movement of heat in the ocean that results in air-sea heat exchange (a diabatic process). However, MHT does not necessarily needs to be tied to air-sea heat exchange or other diabatic processes. We agree that the meridional movement of the subsurface warm blob shielded from the atmosphere as described by the reviewer only contributes to the temporal variability but not the time mean of MHT. Nevertheless, temporal variability of MHT is important to the study of the temporal change of regional oceanic heat content (e.g., to the north or south of the transect). However, it is true that the scenario described by the reviewer will not have a lasting impact on the temperature budget beyond the timescale of the original advection of the blob. Often in quantifying MHT we are interested in the effects of the time-mean circulation, as well as processes with spatial/temporal impacts that are rectified to larger/longer timescales than the original process. We highlight these rectified impacts by showing mostly mesoscale temperature fluxes that have been basinintegrated (Figures 4, 5, 9), or smoothed with a zonal filter (Figures 6c, 8a) to remove
local, essentially rotational fluxes that have no rectified impact at larger scales.

This is described in Section 5.1: "Moreover, rather than being defined relative to a time mean over an arbitrary time period (e.g., the post-spinup time span of the model simulation), the mesoscale MHT is defined as the deviation from a regional background state at each time. Since the integrated volume flux associated with the mesoscale velocity is small across zonal scales » 10° longitude, the MTF smoothed over large zonal scales approximates a heat transport regionally and at each time." (lines 385-389)

2. It might be more natural to analyse the heat transport along a mean streamline rather than across 40N, especially considering the model grid isn't latitude-longitude. This could remove some of the difficulties associated with steady meanders in the model Gulf Stream with scales comparable to those of the "mesoscale" and simplify the physical interpretation of the results (e.g., the cancellations between poleward and equatorward heat transport in Fig. 5a). I understand this isn't trivial to do but if the authors can do it within a reasonable amount of time it would add to the quality of the analysis.

The analysis suggested by the reviewer would be interesting and important, particularly in order to assess the impact of mesoscale fluxes on cross-stream gradients and the Gulf Stream circulation. It would require a substantial amount of time however to refine the method to assess mesoscale fluxes across a streamline and extract the needed output. It is beyond the scope of this paper, but we hope that this analysis is carried out and reported in another manuscript. The purpose of our paper is to investigate to what extent the "eddy" MHT presented in the literature based on temporal decomposition can represent mesoscale variability based on the spatial scales that are usually used to define mesoscales.

OSD
Minor comments:

1. The length of the integration should be mentioned in the text (I only had a hint of it by looking at the period covered by the timeseries displayed –I couldn't see this information in the model description).

This information has been added to the text in the beginning of Section 2.1: "The model integration was started from 15 years of spin-up using CORE normal-year forcing (Large and Yeager 2004), and run over 33 years (corresponding to forcing for the years 1977-2009); our analysis encompasses 32 years and begins in 1978 to focus on the period after the transition from spin-up." (lines 80-82)

2. line 290: "upgradient heat transport" needs to be stated with caution. You might need to remove the rotational component of the heat transport vector first.

We have added the caveat about the rotational component to the text in Section 4.2: "...the large-scale meridional temperature gradient has the same sign here as elsewhere (Fig. 11d). This would imply an upgradient flux of temperature at  $60^{\circ}$ - $50^{\circ}$ W, though it can not be determined solely from this zonal transect whether this apparent upgradient diffusivity is associated with a rotational component of the temperature flux (e.g., Marshall and Shutts, 1981)." (lines 342-345)

3. Abstract, line 5 "...that are more fundamental to the physics of ocean eddies". I am not sure the spatial scales are more fundamental than the temporal scales so maybe this could be removed.

The sentence has been revised as follows: "However, previous analyses of "eddy" MHT in the region have mostly focused on the contributions of time-variable velocity and temperature, rather than considering the association of MHT with distinct spatial scales within the basin." (lines 3-5)

---

## Author Comment (AC2) · 17 Jun 2020

Note: The original reviewer comments are indicated in italic font below, with the author responses and manuscript edits following in non-italic font.

This paper is well written and I recommend publication after some revision/addition.

Thanks to the reviewer for the very thoughtful, helpful comments on this manuscript. We have addressed the reviewer's comments point by point below.

The paper points to its novel aspect a being a new way to do the separation of the eddy vs. large-scale contributions, and primarily as a different view than the Hall and Bryden (1982) separation of the baroclinic component of the heat transport. As such, I think the authors should directly compare some of the computations and maps to the Hall and Bryden method.

For example, in Figure 3, would the HB82 eddy term look different? Same in Figure 4, etc. Jayne and Marotzke (Rev. of Geophys. 2001) did some comparisons of the HB82 decomposition vs. the other time-varying heat transport terms.

A key difference between our decomposition method and that used in HB82 is that HB82 first used a depth average to separate barotropic and baroclinic components of the temperature flux. Later in HB82, a zonal average is used to decompose the baroclinic component further; we use the zonal average first to separate the overturning and zonal-deviation components. We have computed HB82's barotropic and baroclinic components, and the baroclinic zonal-deviation component (what they call the eddy flux) from the POP output. Time mean and interannual/decadal standard deviations of the HB82 eddy flux are shown alongside the mesoscale and time-deviation temperature fluxes in Figure 15. We have described the results of this analysis in Section 5.2, for example:

"To the south and north of this active mesoscale region, all of the "eddy" formulations have much lower time-mean values, with the exception of the baroclinic eddy term which peaks as high as 0.35 PW at 36°N. However, the definition of the baroclinic eddy flux includes large-scale gyre flows that have a baroclinic component, and the baroclinic eddy contribution is generally comparable to or smaller than the large-scale contribution to time-mean MHT (Figure 4a)." (lines 414-418)

Figure 5 seems to indicate that the separation between their large-scale vs. mesoscale is not very great. That is the spatial filter they used doesn't seem to really separate the

spatial scales well, and the spectra in Figure show there isn't really a strong scale separation, especially in the temperature. It should be commented on.

Figure 5 (now Figure 6 in the revised manuscript) does not really provide an indication of the how well the spatial filter separates the large scale and mesoscale v and T. This is because the figure shows the zonally-smoothed product of v and T, which in the case of the mesoscale component is a rectified flux of the mesoscale v and T onto larger scales. In the revised manuscript, the new Figure 3 shows the v and T decomposition along several transects, to show the scale separation between mesoscale and large-scale explicitly. The reviewer correctly notes that the scale separation is not as distinct in temperature, and the temperature spectra is also more red-shifted towards larger scales (Figure 2). However, this is not an impediment to generating substantial mesoscale temperature fluxes when temperature anomalies are advected by mesoscale velocities.

Some enhancements to our method near boundaries have improved the physical interpretation of our results; the benefit is that the cumulative mesoscale volume transport in a transect (as well as across distances »  $10^{\circ}$  longitude within transects) is near zero. Figure 3 shows that the large-scale velocity field preserves the large-scale volume transport (i.e., the barotropic streamfunction), and therefore represents features such as the Gulf Stream as a coarse-resolution model might represent them. The other examples of large-scale and mesoscale temperature flux structure in Figure 3 (at  $28^{\circ}$ N and  $34^{\circ}$ N) illustrate further how this decomposition can diagnose the contributions of large-scale velocity and temperature structure, as described in Section 3.1:

"Most of the non-overturning temperature flux is associated with the large-scale component at both latitudes; however, the mesoscale temperature flux (MTF) switches sign from negative at 28°N to positive at 34°N (Fig. 3d,h). The reason for this is the temperature difference between the core of the northward boundary current and the southward recirculation  $2^{\circ}$  to the east. At 351 m depth (a representative depth for lateral temperature gradients in the thermocline), the temperature at 28°N is lower in the boundary current than it is in the interior recirculation (Fig. 3b), as isopycnals tilt upward sharply approaching the Florida coast. However, at 34°N the temperature peak along the zonal profile is coincident with the boundary current (Fig. 3f), and the temperature peak also has more of a mesoscale signature that explains why the vMTM contributes the most to the MTF (Fig. 3h)." (lines 204-212)

---

## Author Response (AR2)

**Authors' Response to Editor Comments**

Comments to the Author:
Dear Authors,

I am happy with your reply to the reviewers comments and recommend publication subject to considering my suggested minor revisions.

Thanks to the editor for the review and suggested revisions. We have implemented most of the changes, as explained below.

Non-public comments to the Author:
Line 217-219. Suggest modifying "Most of the non-overturning temperature flux is associated with the large-scale component at both latitudes; however, the mesoscale temperature flux (MTF) switches sign from negative at 28oN to positive at 34oN (Fig3d,h)."
TO
"Most of the non-overturning temperature flux is associated with the large-scale component at both latitudes; however, the net mesoscale temperature flux (MTF) switches sign from negative at 28oN to positive at 34oN (Fig3d,h)."

The change has been implemented as suggested. (line 205)

Line 248 delete "overall"

The change has been implemented as suggested. (line 234)

Line 261 write as either '...... interannual/decadal .....' or '....ID ......' as ID previously defined.

As the acronym ID was defined in the previous paragraph, we decided to use just the acronym "ID" in place of "interannual/decadal". (line 247)

Line 265 Check sub-section numbering from this point and onwards. Should this be 4.1 or 3.2?

There were some inconsistencies in the section numbering of the "tracked changes" version of the manuscript, due to the reorganization of the revised manuscript. The section numbering should now be correct in both versions.

Line 423 should this be '<<10o longitude"

The phrase was correct as written, ">>10° longitude", since the sentence is referring to the integrated volume flux associated with the mesoscale being generally negligible over larger scales (>>10° longitude). (line 389)

Figure 1. The gray line indicating 40oN is hard to notice from the black grid lines. A white or other color bold line is suggested.

The authors agree, and the gray lines at 40°N have been changed to white lines; this has also been noted in the figure caption.

Odd wording "(c, d) Same as (c, d) but for surface eddy kinetic energy (EKE); POP
sea surface height is low-pass filtered to remove variability at scales smaller than 0.5_," Suggest change to
"Surface eddy kinetic energy (EKE) (c) altimetry and (d) POP with the sea surface height is low-pass filtered to remove variability at scales smaller than 0.5 …."

The change has been implemented essentially as suggested; the text in the Figure 1 caption now reads:

"Comparison of surface eddy kinetic energy (EKE) from (c) altimetry and (d) POP with the sea surface height low-pass filtered to remove variability at scales smaller than 0.5°…"

Figure 3. In the caption it may also be appropriate to mention that discontinuous curves in decomposition at 351 m show position of land.

Text has been added at the end of the Figure 3 caption to explain this: "In the upper panels, discontinuities in the curves indicate land areas."

[revised manuscript text omitted]